# Mold and Yeast-Like Fungi in the Seaside Air of the Gulf of Gdańsk (Southern Baltic) after an Emergency Disposal of Raw Sewage

**DOI:** 10.3390/jof7030219

**Published:** 2021-03-17

**Authors:** Małgorzata Michalska, Monika Kurpas, Katarzyna Zorena, Piotr Wąż, Roman Marks

**Affiliations:** 1Department of Immunobiology and Environment Microbiology, Faculty of Health Sciences with Institute of Maritime and Tropical Medicine, Medical University of Gdańsk, 80-210 Gdańsk, Poland; monika.kurpas@gumed.edu.pl (M.K.); kzorena@gumed.edu.pl (K.Z.); 2Department of Nuclear Medicine, Faculty of Health Sciences with Institute of Maritime and Tropical Medicine, Medical University of Gdańsk, 80-210 Gdańsk, Poland; piotr.waz@gumed.edu.pl; 3Institute of Marine and Environmental Sciences, University of Szczecin, 70-453 Szczecin, Poland; roman.marks@usz.pl

**Keywords:** emergency disposal of raw of sewage, seaside air, bioaerosol, mold, yeast-like fungi

## Abstract

The aim of this study was to determine the correlation between the meteorological factors and the number of molds and yeast-like fungi in the air in the five coastal towns in the years 2014–2017, and in 2018, after emergency disposal of raw sewage to the Gdańsk Gulf. In the years 2014–2018, a total number of 88 air samples were collected in duplicate in the five coastal towns of Hel, Puck, Gdynia, Sopot, and Gdańsk-Brzeźno. After the application of the (PCA) analysis, this demonstrated that the first principal component (PC1) had a positive correlation with the water temperature, wind speed, air temperature, and relative humidity. The second principal component (PC2) had a positive correlation with the relative humidity, wind speed, wind direction, and air temperature. In 2018, potentially pathogenic mold and yeast-like fungi (*Candida albicans*, *Stachybotrys chartarum* complex, *Aspergillus* section *Fumigati*) were detected in the seaside air. While the detected species were not observed in the years 2014–2017. We suggest that it is advisable to inform residents about the potential health risk in the event of raw sewage disposal into the water. Moreover, in wastewater treatment plants, tighter measures, including wastewater disinfection, should be introduced.

## 1. Introduction

Biological aerosols are a subset of atmospheric particles consisting of both living and non-living organisms, including bacteria, viruses, pollens, molds, yeast-like fungi, and their metabolic products (e.g., mycotoxins) [1,2]. Most molds are not harmful to humans, however, previous studies, including ours, have shown that yeast-like fungi and spores of mold fungi are the etiological factors of many diseases, including allergy, pneumonia, bronchitis, neoplastic diseases, and type 1 diabetes [3,4,5,6]. The storage and sorting of organic waste, composting, agricultural production, food processing, and wastewater treatment systems emit large volumes of bioaerosols, which lead to significant exposure to biological factors [7,8,9]. Studies have demonstrated that not only emergency and uncontrolled wastewater disposals but also outflows from treatment plants and heavy rains cause an increase in the number of microorganisms, such as mold and yeast-like fungi in coastal seawater and sand [8,9,10,11,12]. Moreover, the mold and yeast-like fungi survive in salt seawater and in beach sand for many months [13,14,15]. More than 30 years ago, Anderson (1979) discovered that pathogenic fungi, for instance, *Trichosporon cutaneum, Candida albicans, Microsporum gypseum,* and *Trichophyton mentagrophytes*, could survive in the sand for over a month [16]. A similar study was conducted by other authors who demonstrated that several species of dermatophytes (*Epidermophyton floccosum, Microsporum canis, M. gypseum, T. mentagrophytes, T. rubrum*), and *Scopulariopsis brevicaulis* can survive in the sand from 25 to 360 days [17,18]. Therefore, Vogel et al. suggested that pathogenic yeast-like fungi found in seawater, sewage and beach sand could be a good additional mycological indicator in assessing the safety of marine bathing waters [18].

In terms of mycology, clean air and seawater are of key importance to the health of seaside town inhabitants. This is especially true of holiday resorts that dispose of treated sewage into seawaters. A typical example is seaside bathing areas located along the Gulf of Gdańsk—a bay in the south-eastern part of the Baltic Sea, located between Poland and Russia. Wastewater from the two largest sewage treatment plants is disposed of into the Gulf of Gdańsk [19]. The first (Gdańsk-Wschód) disposes of sewage into the Gulf of Gdańsk to a distance of 2.5 km from the shoreline through a deep water collector, the second (Gdynia-Dębogórze) uses a collector over a distance of 2.3 km to dispose of wastewater to the Bay of Puck (western part of the Gulf of Gdańsk). What is more, from the 15^th^ to 18^th^ May 2018, there was an emergency raw sewage disposal into the Motława River, which flows into the Gulf of Gdańsk (Figure 1). We have adopted a research hypothesis that emergency disposal of raw sewage into the Gulf of Gdańsk could have led to microbiological contamination of the water in the Gulf of Gdańsk and the local air. The preliminary results of our study showed an increase in the number of coliform bacteria and *Escherichia coli* in the seawater and air as a result of emergency disposal of raw sewage to the Gulf of Gdańsk in 2018 [7]. Thus far, the results of mycological tests performed after the disposal have not been presented. Therefore, the aim of this study was to determine the correlation between the meteorological factors and the number of molds and yeast-like fungi in the air in the five coastal towns in the years 2014–2017, and in 2018, after emergency disposal of raw of sewage to the Gdańsk Gulf.

## 2. Materials and Methods

### 2.1. Collection of Air Samples of the Gulf of Gdańsk

In the years 2014–2018, a total of 88 air samples were collected in duplicate, in 5 coastal towns, i.e., Hel, Puck, Gdynia, Sopot, and Gdańsk-Brzeźno on the Gulf of Gdansk (Figure 1). In the years 2014–2017, 62 air samples were collected between 14th May and 14th July, every 28 days between 9:00 a.m. and 2:00 p.m. In 2018 after the emergency disposal of raw sewage, the 26 air samples were collected between 14th May and 23rd July, every 14 days between 9:00 a.m. and 2:00 p.m. The air samples were not collected in the rain and heavy rainfall.

Air samples were collected at a height of 50 cm and about 100 cm from the waterline. In all cases, the samples were collected for 10 min over the Gulf of Gdansk in the 5 coastal towns of Hel, Puck, Gdynia, Sopot, and Gdańsk-Brzeźno. The air samples were collected by impaction with a SAS Super ISO 100 (Milan, Italy) sampler. The nozzle of the sampler was positioned perpendicular to the wind direction. The sampler automatically collected 100-L samples of air. The extracted air was then transported through small holes to a head with a Petri dish containing a Sabouraud dextrose agar medium. The maximum efficiency of the collection was for particulate matter of d50 = 2–4 μm. The flow rate was 90 Lpm. All removable parts of the air sampler were sterilized by autoclaving before sampling, and the sterilized sampler head was cleaned between samples with 70% ethanol.

### 2.2. Mould Fungi Incubation and Identification

The fungi were counted after a 120-hour incubation at 28 °C on Sabouraud Dextrose Agar medium by Merck (Darmstadt, Germany). Yeast-like fungi were identified by CHROMagar Candida, Graso Biotech (Starogard Gdańskicity, Poland).

Mold fungi were identified based on their macro- and microscopic features, with the use of a Nicon Eclipse E2000 microscope at 400, 600, 1000× magnification and a key for the identification of fungi [20,21]. Mould colonies were identified on the basis of the color, texture, topography of the culture surface, smell of the colony, color of the reverse of the colony, and the presence of the diffuse pigment. Microscopic features of the fungal colonies were identified based on their microscopic features, i.e., the presence of macroconidia and microconidia, their shape, and appearance [20,21].

### 2.3. Sample Analysis of Mould Fungi

The number of colonies of fungi were expressed as a colony-forming unit (CFU) per 1 m^3^ of the air (CFU/m^3^). When applying the impact method, we used the Feller table attached to the manual of the air sampler [5,22].

The colonies collected should be revised by the equation: Pr = N[1/N + 1/N − 1 + 1/N − 2 + 1/N − r + 1](1)
where Pr is the revised colony in stage, N is the number of sieve pores, and r is the number of viable particles counted on the agar plate.

The number of colonies of fungi (CFU/m^3^) was calculated using the following equation:C(CFU/m^3^) = T × 1000t(min) × F(L/min)(2)
where C—airborne fungi concentration; CFU—colony-forming unit; T—total colonies after application of the Pr statistical correction; t—sampling time and F—airflow rate.

### 2.4. Characterisation of Meteorological Conditions

During the collection of air samples between spring and summer during 2014–2017 and in 2018, we recorded air and water temperature, humidity, wind speed, and wind direction using a GMH 3330 thermo-hygrometer by Greisinger (Remscheidcity, Germany). The air temperature during 2014–2017 ranged from 26 °C to 3 °C (spring season) and from 20 °C to 16 °C (summer season), respectively. Relative humidity in the spring season was between 30% and 88%, and from 59% to 82% in the summer season. Wind speed in the spring season varied between 0 and 31 km/h and between 7 and 25 km/h in the summer. The air temperature in 2018 fluctuated between 27 °C and 10 °C in the spring and between 27 °C to 15 °C in the summer. Relative humidity in the spring season was between 39% and 93%, and from 44% to 70% in the summer season. Wind speed in the spring season varied between 0 and 15 km/h and between 2.6 and 32 km/h in the summer. Air samples were not collected when it rained.

### 2.5. Statistical Analysis

To search for hidden relationships and regularities between meteorological factors and the number of mold and yeast-like fungi in the air, we used one of the numerous methods of factor analysis. For data analysis, Principal Component Analysis (PCA) ready-made procedures from the “ggfortify”, “FactoMineR,” and “factoextra” packages [23,24,25,26] were used. The algorithm of the PCA method allowed the data to be transformed (using the features of the eigenvalues and eigenvectors of the covariance matrix or the correlation matrix for this purpose) in order to obtain the greatest possible differences in standard deviations in the new variables. The first major component of PC1 is related to most of the variability of the original data set, and the second major component of PC2 is related to the second largest component, and so on [27]. Assuming that the first few components contained a significant amount of variation in the original data set, together they may account for almost all of the variability in the data and thus simplify the interpretation of the results. The obtained results of the statistical analysis made it possible to determine the correlation between meteorological factors, the number of molds, the number of yeast-like fungi in the coastal air in the researched locations on the Gulf of Gdańsk, and PC1 and PC2. The PCA statistical analysis was performed for 2 research periods, 2014–2017 and 2018.

## 3. Results

### 3.1. The Principal Component Analysis of Mold and Yeast-Like Fungi Detected in Air Samples in the Five Seaside Towns in the Study Period of 2014–2017 and in 2018

The first three main principal components explained almost 78% of the total variance. The first principal component (PC1) explained 33.54% of the total variance. The second principal component (PC2) explained 27.23% of the total variance. The third principal component (PC3) explained 17.15% of the variance (Table 1). The PCA loading plot of the first two principal components compared the numbers of mold and yeast-like fungi and the meteorological factors in the coastal towns of Hel, Puck, Gdynia, Sopot, and Gdańsk-Brzeźno in the years 2014–2017, and in 2018 (Figure 2).

Table 2 shows the values of the correlation between the variables used in the model and the main components shown in Figure 2. For each of the determined correlation values, the *p*-value was given. Table 2 includes only those variables that, at the assumed significance level, gave a statistically significant result *p* < 0.00001.

The first principal component (PC1) had a positive correlation (*p* < 0.00001) with the water temperature, wind speed, air temperature, and relative humidity, explaining 33.54% of the total variance observed. The second principal component (PC2) had a positive correlation (*p* < 0.00001) with the relative humidity, wind speed, and wind direction, and a negative correlation with the air temperature, explaining 27.23% of the variance (Table 2).

### 3.2. The Correlation between the Number of Mold Fungi Aspergillus sp., Penicillium sp., Cladosporium sp. in the Research Period of 2014–2017 and in 2018

In the next stage of the study, statistical analysis based on PCA was used to evaluate the number of mold fungi *Aspergillus* sp. (Table 3 and Table 4), *Penicillium* sp., (Table 5 and Table 6) and *Cladosporium* sp. (Table 7 and Table 8) in the years 2014–2017 and in 2018.

#### 3.2.1. The Correlation between the Number of Mold Fungi Aspergillus sp., the Meteorological Elements in the Research Period of 2014–2017 and in 2018

The analysis of the PCA eigenvalues of the correlation matrix (Table 3) revealed the three main principal components, which could explain 79.16% of the total variance. The first principal components (PC1) explained 33.10% of the total variance. The second principal component (PC2) explained 27.49% of the total variance. The third principal component (PC3) explained 18.57% of the variance.

The PCA loading plot of the first two principal components comparing the numbers of *Aspergillus* sp. and the meteorological factors in the five coastal towns in the years 2014–2017, and in 2018 (Figure 3).

Table 4 shows the values of the correlation between the variables used in the model and the principal components shown in Figure 3, along with the *p*-value. Table 4 presents only those variables that, at the assumed significance level, gave a statistically significant result *p* < 0.00001.

The first principal component was a significant correlation (*p* < 0.00001) with the water temperature, wind speed, and air temperature, explaining 33.10% of the total variance observed (Table 4). The second principal component (PC2) was correlated with the relative humidity, wind speed, and air temperature explaining 27.49% of the variance.

#### 3.2.2. The Correlation between the Number of Mold Fungi Penicillium sp. the Meteorological Elements in the Research Period of 2014–2017 and in 2018

Analysis of the eigenvalues of the correlation matrix (Table 5) revealed the three main principal components, which could explain 77.14% of the total variance. The first principal components (PC1) explained 33.20% of the total variance. The second principal component (PC2) explained 27.79% of the total variance. The third principal component (PC3) explained 16.16% of the variance.

The PCA loading plot of the first two principal components comparing the numbers of *Penicillium* sp. and the meteorological factors in the five coastal towns in the years 2014–2017, and in 2018 (Figure 4).

Table 6 presents the values of the correlation between the variables used in the model and the main components presented in Figure 4. For each of the determined correlation values, the *p*-value was given. Table 6 contains only those variables which, at the assumed significance level, gave a statistically significant result.

The first principal component (PC1) was the combination of the wind speed and water temperature and then relative humidity and air temperature, explaining 33.10% of the total variance observed (Table 6). The principal component (PC2) was correlated with the air temperature, relative humidity, water temperature (*p* < 0.00001), and wind direction (*p* = 0.00016), explaining 27.49% of the variance.

#### 3.2.3. The Correlation between the Number of Mold Fungi Cladosporium sp. and the Meteorological Factors in the Research Period of 2014–2017 and in 2018

The PCA analysis of the eigenvalues of the correlation matrix (Table 7) revealed the three main principal components, which could explain 78.2% of the total variance. The first principal components (PC1) explained 32.76% of the total variance. The second principal component (PC2) explained 28.42% of the total variance. The third principal component (PC3) explained 17.0% of the variance.

Table 8 presents the values of the correlation between the variables used in the model and the main components presented in Figure 5. For each of the determined correlation values, the *p*-value was given. Table 8 lists only those variables that, at the assumed significance level, gave a statistically significant result.

The most important principal component (PC1) of mold fungi *Cladosporium* sp. was the significant correlation (*p* < 0.00001) with the water temperature, wind speed, and air temperature, explaining 32.76% of the total variance observed (Table 8). The second principal component (PC2) was the significant correlation with the relative humidity, air temperature, and wind speed, explaining 28.42% of the variance.

### 3.3. The Qualitative Assessment of Mold and Yeast-Like Fungi in the Atmospheric Air of the Seaside Towns in the Years 2014–2017

In the air samples collected in the years 2014–2017 in the towns of Hel, Puck, Gdynia, Sopot, and Gdańsk-Brzeźno, *Ascomycota* (98.29%), *Basidiomycota* (1.52%), and *Zygomycota*, (*Fungi Incertae Sedis*) were detected (0.19%) (Figure 3).

Three classes of *Ascomycota* fungi were found—*Eurotiomycetes* (77.50%), *Dothideomycetes* (18.44%), and *Saccharomycetes* (2.34%). In the *Eurotiomycetes* class, the following genus was isolated: *Penicillium* (63.24%), *Aspergillus* (13.56%), and *Trichophyton* (0.70%). Within the *Penicillium* genus, *Penicillium* section *Viridicata* (40.11%) and *Penicillium* section *Chrysogena* (23.13%) were detected. Within the *Aspergillus* genus, there was *Aspergillus* section *Nigri* (13.56%). Within the *Trichophyton* genus, the *Trichophyton mentagrophytes* complex (0.70%) was isolated. In the class of *Dothideomycetes*, the *Cladosporium* genus was found, with the *Cladosporium herbarum* complex (16.79%) and the *Aureobasidium* genus, with the *Aureobasidium pullulans* complex (1.65%). In the *Saccharomycetes* class, the *Saccharomycetes* genus was isolated (2.34%). In the *Basidiomycota* phylum, the *Cystobasidiomycetes* class, the *Rhodotorula* genus, with the *Rhodotorula* sp. (1.52%) was detected. In the *Zygomycota* phylum, the *Mucoromycotina* class, with the *Mucor mucedo* group (0.19%) was detected. The percentage of mold and yeast-like fungi in the seaside air is shown in Figure 6.

### 3.4. The Qualitative Assessment of Mold and Yeast-Like Fungi in the Air of Seaside Towns in 2018

*Ascomycota* (96.17%), *Basidiomycota* (1.21%), and *Zygomycota* (*Fungi Incertae Sedis*) (2.62%) were detected in the air samples collected in 2018 in the seaside towns of Hel, Puck, Gdynia, Sopot, and Gdańsk-Brzeźno (Figure 4). In the *Ascomycota* phylum, four classes of fungi were found: *Eurotiomycetes* (60.07%), *Dothideomycetes* (4.60%), *Saccharomycetes* (27.26%), and *Sordariomycetes* (4.24%).

The genus isolated within the *Eurotiomycetes* class were: *Penicillium* (20.72%), *Aspergillus* (38.5%), and *Trichophyton* (0.85%). Within the *Penicillium* genus, *Penicillium* section *Viridicata* (11.87%) and *Penicillium* section *Chrysogena* (8.84%) were detected. Within the *Aspergillus* genus, the species: *Aspergillus* section *Nigri* (37.05%) and *Aspergillus* section *Fumigati* (1.45%) were isolated. Within the *Trichophyton* genus, the *Trichophyton mentagrophytes* complex was detected (0.85%). In the *Dothideomycetes* class, there was *Cladosporium* genus, the *Cladosporium herbarum* species complex (4.31%), and the *Alternaria* genus with the *Alternaria alternata* complex (0.29%). In the *Sordariomycetes* class, the *Stachobytris* genus with the *Stachybotrys chartarum* complex (4.24%) was found. In the *Saccharomycetes* class, the *Candida* genus was isolated with *Candida albicans* (27.26%). In the B*asidiomycota* phylum, the *Cystobasidiomycetes* class, the *Rhodotorula* genus, and the *Rhodotorula* sp. (1.21%) were detected. In the *Zygomycota* phylum, the *Mucoromycotina* class with the *Mucor mucedo* group (2.62%) was found. In 2018, potentially pathogenic and allergenic mold and yeast-like fungi were detected in the seaside air, such as *A.* section *Fumigati* (1.45%), *S. chartarum* complex (4.24%), and *C. albicans* (27.26%). The species were not observed in the years 2014–2017. The percentage share of these mold and yeast-like fungi in the samples of seaside air is shown in Figure 7.

## 4. Discussion

The analysis based on the PCA correlation of mold and yeast-like fungi detected in air samples in the seaside towns and meteorological factors showed a statistically significant relationship (*p* < 0.00001). The principal component (PC1) was correlated with the air and water temperature, and wind speed explains 33.54% of the total variance. The second principal component (PC2) was correlated with the relative air humidity, air temperature, direction, and wind speed, and explains 27.23% of the total variance (Table 2). Similar research results were obtained by Grinn-Gofron and Bosiacka [28]. Their 4-year study showed that air temperature, dew point, relative humidity, and average wind speed had the greatest influence on the composition of spores in the air [28]. Previous interdisciplinary studies too, including our research, showed that the direction and speed of the wind are one of the most important meteorological factors affecting the formation of aerosols at the water-air interface [29,30,31,32]. In our study, the important meteorological factors affecting airborne spore of *Aspergillus* sp. and *Penicillium* sp. concentrations were relative humidity and wind speed. The same results were noted by Grinn-Gofroń in Szczecin. The daily values of relative humidity and average wind speed were positively correlated for *p* = 0.001 and *p* = 0.01, respectively [33]. In addition, the principal component (PC1) and (PC2) of *Cladosporium* sp. was significantly correlated (*p* < 0.00001) with the wind speed, air temperature, and relative humidity. In the same way, air temperature, wind speed, and relative humidity were associated with *Cladosporium* spore dispersal in Morocco [34]. In addition, our PCA statistical study of the seaside air of 2018 revealed that, compared to the number of fungi detected in the years 2014–2017, the highest number of mold and yeast-like fungi, after emergency disposal of sewage into the Gulf of Gdańsk, was detected in Hel, Sopot, and Gdańsk-Brzeźno (Figure 2). We believe that the greater number of mold fungi in the air samples in these seaside towns could have been influenced not only by water and air temperature, wind speed, and humidity but also by the direction of the wind (SE) blowing from the Bay of Gdańsk. We suggest that it is advisable to inform residents about the potential health risk in the event of raw sewage disposal into the water.

On the other hand, in Gdynia, in 2018, neither mold nor yeast-like fungi were found compared to the period of 2014–2017. The lack of mold and yeast-like fungi in the air of Gdynia in 2018 could be caused by the direction of the wind blowing from the sea (NE). The Gulf of Gdańsk is partially separated from the Baltic Sea by the Hel Peninsula and the city of Hel. In Hel, the dominant wind direction is west, and in Gdynia, a north-east direction. In the city of Gdynia, due to the presence of the Hel Peninsula, the wind blowing from the direction of the sea is less frequently observed.

To our knowledge, no results of mycological studies on the quality of seaside air after emergency disposal of raw sewage are available. Therefore, the outcomes of 2018 were compared to the quality of air at the sewage treatment plant in order to indicate the likely origin of the species. Filamentous fungi of the genus *Aspergillus*, *Cladosporium,* and *Mucor* and yeast-like fungi, for example, *Candida,* were detected in domestic human and animal sewage [35,36,37,38]. Michałkiewicz et al. found that the majority of yeast-like fungi isolated from the air of the four wastewater treatment plants was *Candida* [39]. Other studies demonstrated that the following mold and yeast-like fungi were predominant: *Cladosporium* sp., *A. fumigatus, A. alternata, C. albicans,* and *Rhodotorula* sp. [40,41,42,43,44]. Potentially pathogenic fungi, such as *Olpidium, Paecilomyces, Aspergillus, Rhodotorula, Penicillium, Candida, Synchytrium, Phyllosticta,* and *Mucor* have been detected in three wastewater treatment plants located in the Gauteng province of the Republic of South Africa [45]. In Portuguese studies, mold fungi *Aspergillus*, *Fusarium,* and yeast-like fungi *Candida* were found in beach sand contaminated with leaking toilet sewage [12].

In turn, other researchers conducted a sanitary evaluation of sand and water from 16 beaches of São Paulo State, Brazil [46]. Ninety-six samples each of wet and dry sand and seawater were collected and analyzed for fecal indicator bacteria. Correlation analysis indicated a significant relationship between fecal indicator densities in wet sand and seawater. There was a significant correlation between the densities of fecal coliforms and fecal streptococci for both types of sand, and this correlation was higher in wet sand. These data suggest the necessity of some criteria for microbiological control [46]. Transmission of infectious diseases in terrestrial beach environments can occur via direct exposure to microbes found in sand or through the flux of microbes from water to sand within the swash or intertidal zone. In addition to direct exposure, sand can also serve as a vehicle for transferring pathogenic microbes to and from the adjacent water [10].

Recent research suggests that being in and using the beach may be a risk factor for infectious diseases, thus monitoring of both seawater, beach sand, and coastal air is warranted [7,10,46,47].

In conclusion, the study results of 2018 indicate that untreated wastewater associated with emergency disposal to the Gulf of Gdańsk was a likely source of mold and yeast-like fungi in the seaside air. The analysis of PCA data demonstrated a statistically significant relationship between the meteorological factors and the number of mold and yeast-like fungi reported in the period of 2014–2017 and in 2018. Although failures of sewage treatment plant collectors, heavy rainfall, and floods occur quite often, they should not lead to an increase in the number of potentially pathogenic bacteria and mold and yeast-like fungi in the coastal seawater and air. Therefore, it is important to build new wastewater treatment plants, and expand and modernize the existing ones, thus that pathogenic microorganisms are effectively eliminated in the process of wastewater treatment. Moreover, tighter measures, including wastewater disinfection, should be introduced in wastewater treatment plants.

## Figures and Tables

**Figure 1 jof-07-00219-f001:**
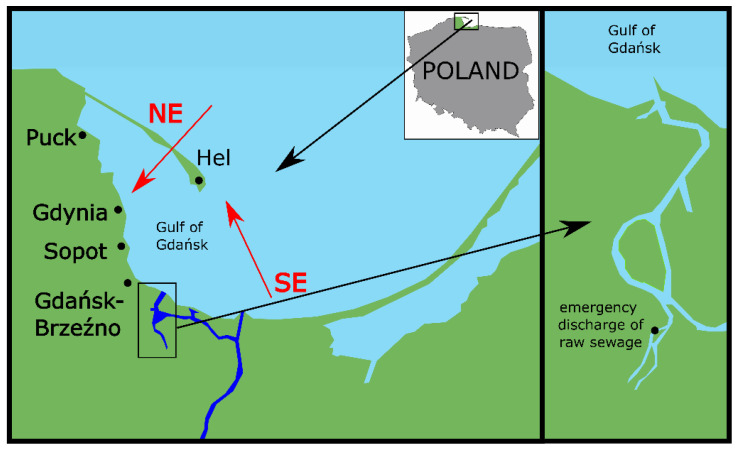
The emergency discharge of raw sewage into the Motława River to the Gulf of Gdańsk in 2018. Map showing the five coastal towns: Hel, Puck, Gdynia, Sopot, and Gdańsk-Brzeźno, where microbiological coastal air were samples collected. The wind direction marked on the map were Southeast (SE) and Northeast (NE) (red arrow). Place of emergency discharge of raw sewage into the Motława River to the Gulf of Gdańsk (black arrow).

**Figure 2 jof-07-00219-f002:**
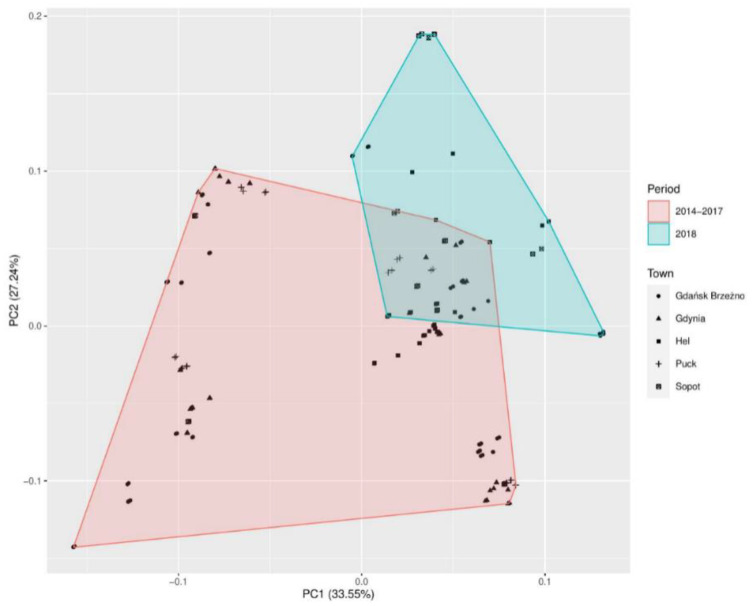
Principal component analysis (PCA) of mold and yeast-like fungi detected in air samples in the seaside towns and meteorological factors. The plot score of the first two principal components contains almost 60.78% of the explained variance. Two clusters can be distinguished: The red one is composed in the study period of 2014–2017 and the blue cluster in 2018.

**Figure 3 jof-07-00219-f003:**
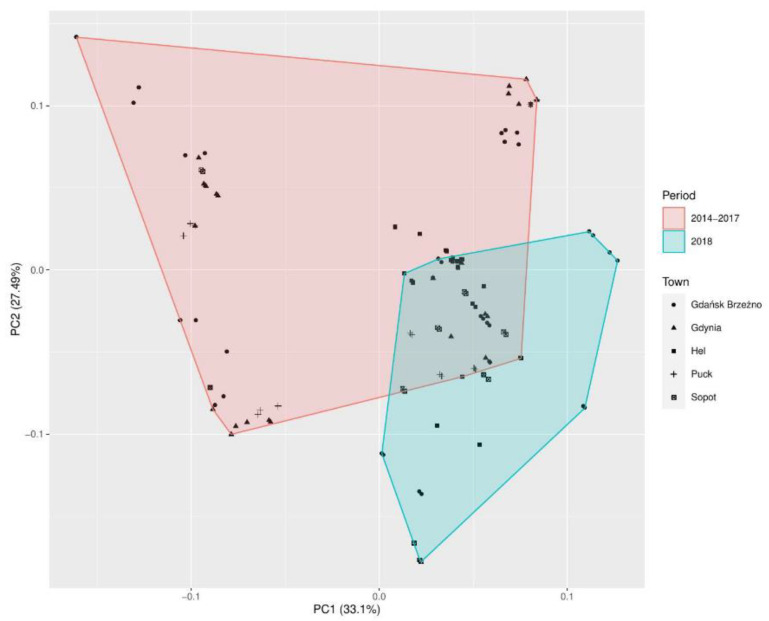
Principal component analysis (PCA) of *Aspergillus* sp. detected in air samples in the seaside towns (Hel, Puck, Gdynia, Sopot, and Gdańsk Brzeźno). The plot score of the first two principal components contained almost 60.59% of the explained variance. Two clusters can be distinguished: The red one is composed in the study period of 2014–2017 and the blue cluster in 2018.

**Figure 4 jof-07-00219-f004:**
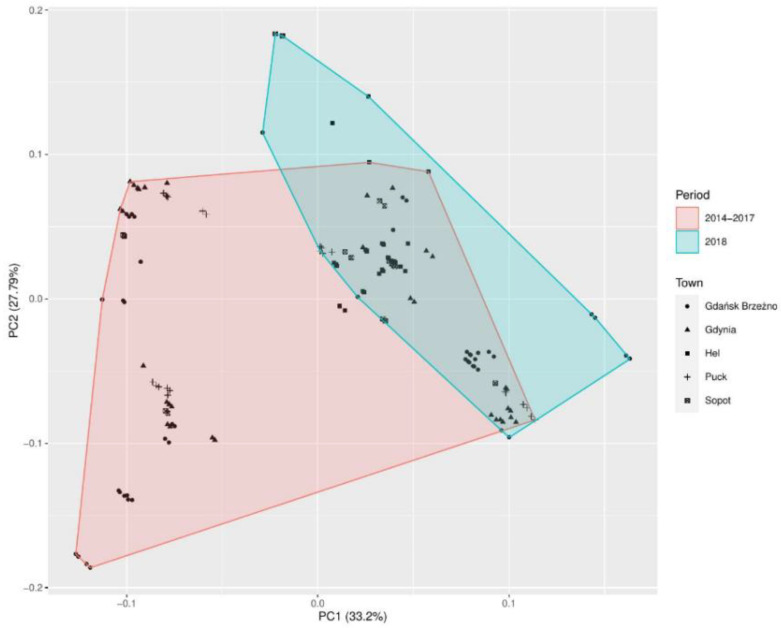
Principal component analysis (PCA) of *Penicillium* sp. detected in air samples in the seaside towns (Hel, Puck, Gdynia, Sopot, and Gdańsk Brzeźno). Here the plot score of the first two principal components is reported. It contains almost 60.98% of the explained variance. Two clusters can be distinguished: The red one is composed in the study period of 2014–2017 and the blue cluster in 2018.

**Figure 5 jof-07-00219-f005:**
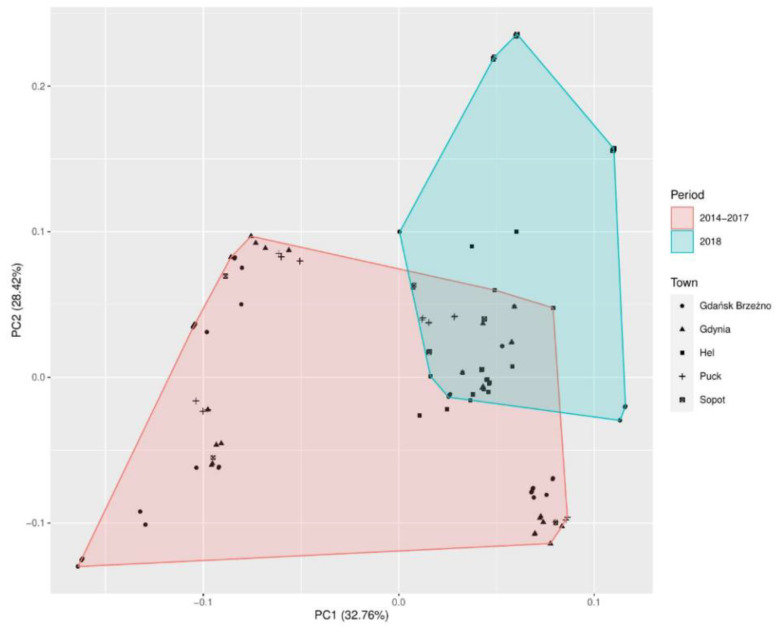
Principal component analysis (PCA) of *Cladosporium* sp. detected in air samples in the seaside towns (Hel, Puck, Gdynia, Sopot, and Gdańsk Brzeźno). The plot score of the first two principal component contains almost 61.17% of the explained variance. Two clusters can be distinguished: The red one is composed in the study period of 2014–2017 and the blue cluster in 2018.

**Figure 6 jof-07-00219-f006:**
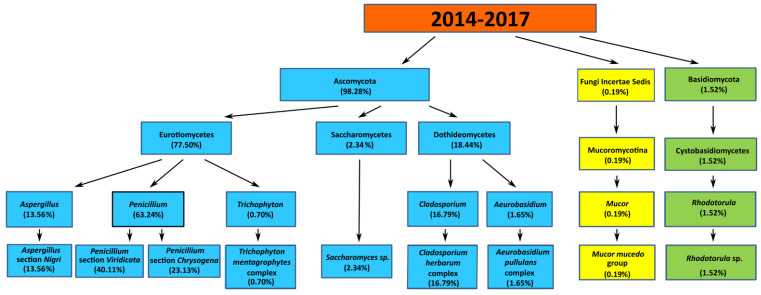
The percentage share of mold and yeast-like fungi in the air of the seaside towns (Hel, Puck, Gdynia, Sopot, and Gdańsk-Brzeźno) in the years 2014–2017.

**Figure 7 jof-07-00219-f007:**
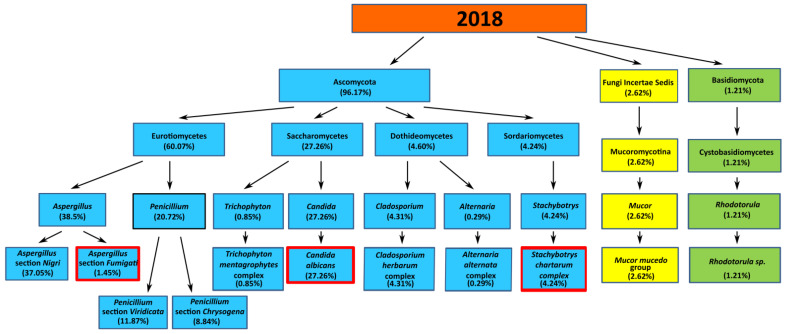
The percentage share of mold and yeast-like fungi in the air of seaside towns (Hel, Puck, Gdynia, Sopot, and Gdańsk-Brzeźno) in 2018. The red frame marks the species were not observed in the years 2014–2017.

**Table 1 jof-07-00219-t001:** Eigenvalues and values of the explained variance for the model from the mold and yeast-like fungi license.

	Eigenvalue	Explained Part of Multivariate Variability of Accessions [%]	Cumulative Part of Multivariate Variability [%]
PC1	2.0127	33.5458	33.5458
PC2	1.6341	27.2358	60.7816
PC3	1.0294	17.1567	77.9383
PC4	0.8123	13.5377	91.4760
PC5	0.2931	4.8847	96.3607
PC6	0.2184	3.6393	100.0000

**Table 2 jof-07-00219-t002:** The values of the correlation between PC1, PC2, and the variables used to build the model.

Variables	PC1		PC2	
	Correlation	*p*-Value	Correlation	*p*-Value
water temperature (°C)	0.8220	<0.00001	−0.3612	<0.00001
wind speed (km/h)	0.7567	<0.00001	0.5178	<0.00001
air temperature (°C)	0.6263	<0.00001	−0.6215	<0.00001
relative humidity %	0.4883	<0.00001	0.8031	<0.00001
the number of mold and yeast-like fungi (CFU/m^3^)	0.3608	<0.00001	−0.2039	0.00665
wind direction	-	-	0.4035	<0.00001

**Table 3 jof-07-00219-t003:** Eigenvalues and values of the explained variance for the model from the *Aspergillus* sp. license.

	Eigenvalue	Explained Part of Multivariate Variability of Accessions [%]	Cumulative Part of Multivariate Variability [%]
PC1	1.9859	33.0988	33.0988
PC2	1.6496	27.4930	60.5919
PC3	1.1141	18.5688	79.1606
PC4	0.7382	12.3037	91.4644
PC5	0.2952	4.9196	96.3840
PC6	0.2170	3.6160	100.0000

**Table 4 jof-07-00219-t004:** The values of the correlation between PC1, PC2, and the variables used to build the model.

Variables	PC1		PC2	
	Correlation	*p*-Value	Correlation	*p*-Value
water temperature (°C)	0.8264	<0.00001	−0.3557	<0.00001
wind speed (km/h)	0.7576	<0.00001	0.5263	<0.00001
air temperature (°C)	0.6393	<0.00001	−0.6195	<0.00001
relative humidity %	0.4825	<0.00001	0.8094	<0.00001
the number of *Aspergillus* sp. (CFU/m^3^)	0.2921	0.00008	−0.2827	0.00014

**Table 5 jof-07-00219-t005:** Eigenvalues and values of the explained variance for the model from the *Penicillium* sp. license.

	Eigenvalue	Explained Part of Multivariate Variability of Accessions [%]	Cumulative Part of Multivariate Variability [%]
PC1	1.9918	33.1967	33.1967
PC2	1.6671	27.7855	60.9822
PC3	0.9693	16.1555	77.1377
PC4	0.8699	14.4989	91.6367
PC5	0.2942	4.9041	96.5408
PC6	0.2076	3.4592	100.0000

**Table 6 jof-07-00219-t006:** The values of the correlation between PC1, PC2, and the variables used to build the model.

Variables	PC1		PC2	
	Correlation	*p*-Value	Correlation	*p*-Value
wind speed (km/h)	0.8688	<0.00001	0.2811	0.00016
water temperature (°C)	0.6987	<0.00001	−0.5693	<0.00001
relative humidity %	0.6545	<0.00001	0.6085	<0.00001
air temperature (°C)	0.4711	<0.00001	−0.7575	<0.00001
the number of *Penicillium* sp. (CFU/m^3^)	0.3130	<0.00001	0.3216	0.00001
wind direction	-		0.4653	<0.00001

**Table 7 jof-07-00219-t007:** Eigenvalues and values of the explained variance for the model from the *Cladosporium* sp. license.

	Eigenvalue	Explained Part of Multivariate Variability of Accessions [%]	Cumulative Part of Multivariate Variability [%]
PC1	1.9655	32.7577	32.7577
PC2	1.7050	28.4173	61.1751
PC3	1.0191	16.9845	78.1596
PC4	0.8001	13.3358	91.4954
PC5	0.2921	4.8681	96.3634
PC6	0.2182	3.6366	100.0000

**Table 8 jof-07-00219-t008:** The values of the correlation between PC1, PC2, and the variables used to build the model.

Variables	PC1		PC2	
	Correlation	*p*-Value	Correlation	*p*-Value
water temperature (°C)	0.8287	<0.00001	−0.3116	<0.00001
wind speed (km/h)	0.7492	<0.00001	0.5426	<0.00001
air temperature (°C)	0.6621	<0.00001	−0.6036	<0.00001
relative humidity %	0.4723	<0.00001	0.7982	<0.00001
the number of *Cladosporium* sp. (CFU/m^3^)	0.2199	0.00336	−0.4275	<0.00001
wind direction			0.3597	<0.00001

## Data Availability

Not applicable.

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
