# Peer review of "Mold and Yeast-Like Fungi in the Seaside Air of the Gulf of Gdańsk (Southern Baltic) after an Emergency Disposal of Raw Sewage"

_jof, 2021, doi:10.3390/jof7030219_

Round 1

Reviewer 1 Report

The article “Mold and yeast-like fungi in the seaside air of the Gulf of Gdansk (Southern Baltic) after an emergency disposal of raw sewage" raises an interesting and important issue of mycological air pollution in coastal regions.

The research was well documented and statistically designed.

In my opinion, it should be published in full 

Author Response

Reviewer 1

Comments for the author The article “Mould and yeast-like fungi in the seaside air of the Gulf of Gdansk (Southern Baltic) after an emergency disposal of raw sewage” raises an interesting and important issue of mycological air pollution in coastal regions.

The research was well documented and statistically designed. In my opinion, it should be published in full.

Author’s Response: Thank you for your very good opinion. It is a great honour for the authors to read such a positive opinion. Thank you once more.

Reviewer 2 Report

Unfortunatelly I cannot encourage publication of manuscript „Mold and yeast-like fungi in the seaside air of the Gulf of  Gdańsk (Southern Baltic) after an emergency disposal of raw sewage“ due to several mainly methodical issues“

There is not exactly mentioned when and where the samples were collected

There are not mentioned any replications

Number of samples seems to be low for high quality study

Identification of fungi based solely on morphological traits is not the state of art technique

CFU values should be tranformed to obtain normal distirbution rather than non-parametic test used

Use of contingency tables has a several assumptions and I am not sure that they were fulfilled according provided data.

The effect of emergency disposal of raw sewage cannot be fully assessed if targeted sampling was not done.

Many of these issues cannot be improved thus the manuscript should be rejected.

Round 2

Reviewer 2 Report

I would thank to authors of manuscript for answers. Some of them explained problematic passages, but others are only excuses.

Idea of article is fine i.e. prove that emergency discharge of sewage probably increased numbers of some yeast and mould species in the air. However authors answers did not change my mind about quality of analysis.

My first comment, how and when was samples collected, resulted to expanded material and methods by new paragraph. It is fine, however I would rather see the table consisting of information when and where sampling was done and how many samples were taken.

This is important in relation of my concerns about low sample numbers (especially in year 2018). Such information is not available in manuscript and requested table will resolve my concerns.

Authors chose mosaic graph for illustration of chi square analysis, but this type of graph can hide the real numbers thus without any information about numbers  (some of them are mentioned in discussion – the part where it should not be) readers cannot say how big is effect.

I cannot consider analysis of CFU data by contingency tables to be reliable despite some researchers may prefer it. I tried to search for similar statistical evaluation and I was not very successful. Moreover authors used average values, but in case of such data as CFU it commonly happens that two samples contain 0, 5 and 1000 CFU and this result to average 335 but such number is has very low reliability (look at table 1).

Despite comparison between samples is not the main aim of study, some methods used in environmental microbiology can better resolve differences between samples and changes of microbial community PCA/PCoA/NMDS, PERMANOVA/Anosim and many others.

It is a fact that morphological identification cannot provide comprehensive classification. Such fact should be at least discussed in article, when solely morphological identification was used (which is not bad, but resolution is low). It is also connected to use of species names. Aspergillus niger cannot by morphologically distinguished among other Asperigillus species like A. tubigensis, A. awamori, A. foetidus …Thus it should be referred as Aspergillus section nigri.

Similarly Trichophyton mentagrophytes form a complex of species which are not distinguishable morphologically and so on.

Discusion should not contain results like in L 311-320

I encourage authors to less discuss negative effects of species like L321-329 and be more focused on their occurrence in environment especially in context of results like in L350-353.

Language quality and preparation of manuscript also have some flaws

In table 1 p-value testu Freidmana comes from Polish

L244 L266 L350 Genii should be genera

Latin names should be used in short form when appears second and next time in text

I would encourage authors to do new statistical analysis, correct English, prepare new manuscript and resubmit it as the topic is interesting.

Author Response

Comments for the author

 Reviewer 2

I would thank to authors of manuscript for answers. Some of them explained problematic passages, but others are only excuses. Idea of article is fine i.e. prove that emergency discharge of sewage probably increased numbers of some yeast and mould species in the air. However authors answers did not change my mind about quality of analysis.

My first comment, how and when was samples collected, resulted to expanded material and methods by new paragraph. It is fine, however I would rather see the table consisting of information when and where sampling was done and how many samples were taken. This is important in relation of my concerns about low sample numbers (especially in year 2018). Such information is not available in manuscript and requested table will resolve my concerns.

Author’s Response: Thank you very much for your opinion and remarks. In the following pages are our point-by-point responses to each of the comments of the reviewer.

In the years 2014-2018 a total of 88 air samples were collected in duplicate, in 5 coastal towns, on the Gulf of Gdansk. In the years 2014-2017, 62 air samples were collected between 14th May and 14th July, every 28 days between 9:00 a.m. and 2:00 p.m.

In 2018 after an emergency disposal of raw sewage 26 air samples were collected between 14th May and 23rd July, every 14 days between 9:00 a.m. and 2:00 p.m.

The "raw data" is available from the corresponding author.

In addition, the number of collected samples is comparable to the number of trials by other authors:

  1. Mayol, E., Jiménez, M. A., Herndl, G. J., Duarte, C. M., & Arrieta, J. M. (2014). Resolving the abundance and air- sea fluxes of airborne microorganisms in the North Atlantic Ocean. Frontiers in Microbiology, 5(OCT). https://doi.org/10.3389/fmicb.2014.00557

Material and methods from the paper by Mayol, E et al., are presented of 31 samples.

In turn, in another publication by Genitsaris, S et al., the authors demonstrated their findings of 27 air samples.

 Genitsaris, S., Stefanidou, N., Katsiapi, M., Kormas, K. A., Sommer, U., Moustaka-Gouni, M., 2017. Variability of airborne bacteria in an urban Mediterranean area (Thessaloniki, Greece). Atmos. Environ. 157, 101–110.

----------------------------------------------------------------------------------------

Authors chose mosaic graph for illustration of chi square analysis, but this type of graph can hide the real numbers thus without any information about numbers  (some of them are mentioned in discussion – the part where it should not be) readers cannot say how big is effect.

Author’s Response: Thank you for your remark. As suggested by the reviewer, "mosaic graph for illustration of chi square analysis" was removed in the revised version of the manuscript. In addition, our collected samples were subjected to another statistical analysis of Principal Component Analysis (PCA) - one of the statistical methods proposed by the reviewer.

----------------------------------------------------------------------------------------

I cannot consider analysis of CFU data by contingency tables to be reliable despite some researchers may prefer it. I tried to search for similar statistical evaluation and I was not very successful. Moreover authors used average values, but in case of such data as CFU it commonly happens that two samples contain 0,5 and 1000 CFU and this result to average 335 but such number is has very low reliability (look at table 1).

Author’s Response: Thank you for your remark. In the revised version of our manuscript, Table 1 was removed along with the description of the average number of mold and yeast-like fungi (CFU/m3). Our data was once again subjected to another statistical analysis of Principal Component Analysis (PCA), L141-158 - one of the statistical methods proposed by the reviewer. The obtained results are presented in the section: Results, L 161-L286.

Despite comparison between samples is not the main aim of study, some methods used in environmental microbiology can better resolve differences between samples and changes of microbial community PCA/PCoA/NMDS, PERMANOVA/Anosim and many others.

Author’s Response: Thank you for your remark. As suggested by Reviewer 2, in the revised version of the manuscript we used a different statistical analysis. In our research we used PCA statistical analysis and recalculated the data. The description of the analysis is presented in Section 3. In the Results section, we presented the calculated data as tables and figures from the L 161 to L286.

----------------------------------------------------------------------------------------

It is a fact that morphological identification cannot provide comprehensive classification. Such fact should be at least discussed in article, when solely morphological identification was used (which is not bad, but resolution is low). It is also connected to use of species names. Aspergillus niger cannot by morphologically distinguished among other Asperigillus species like A. tubigensis, A. awamori, A. foetidus …Thus it should be referred as Aspergillus section nigri. Similarly Trichophyton mentagrophytes form a complex of species which are not distinguishable morphologically and so on.

Author’s Response: Thank you for your remark. As suggested by the Reviewer 2, we have corrected the names of mold fungi, e.g., from Aspergillus nigier to Aspergillus section nigri, Trichophyton mentagrophytes to Trichophyton mentagrophytes complex and so on. Enclosed is chapter 3.4 with the corrected names of the mold fungi.

3.4. The qualitative assessment of mould and yeast-like fungi in the atmospheric air of the seaside towns in discusion should not contain results like in L 311-320

In the air samples collected in the years 2014-2017 in the towns of Hel, Puck, Gdynia, Sopot and Gdańsk-Brzeźno, Ascomycota (98.29%), Basidiomycota (1.52%) and Zygomycota, (Fungi Incertae Sedis) were detected (0.19%) (Figure 3).

Three classes of Ascomycota fungi were found - Eurotiomycetes (77.50%), Dothideomycetes (18.44%) and Saccharomycetes (2.34%). In the Eurotiomycetes class, the following genus were isolated: Penicillium (63.24%), Aspergillus (13.56%) and Trichophyton (0.70%). Within the Penicillium genus, Penicillium section Viridicata (40.11%) and Penicillium section Chrysogena (23.13%) were detected. Within the Aspergillus genus, there was Aspergillus section Nigri (13.56%). Within the Trichophyton genus, the Trichophyton mentagrophytes species complex (0.70%) was isolated. In the class of Dothideomycetes, the Cladosporium genus was found, with the Cladosporium herbarum species complex (16.79%), and the Aureobasidium genus, with the Aureobasidium pullulans species complex (1.65%). In the Saccharomyces class (2.34%), the Saccharomycetes genus was isolated. In the Basidiomycota phylum, the Cystobasidiomycetes class, the Rhodotorula genus, with the Rhodotorula mucilaginosa species (1.52%) was detected. In the Zygomycota phylum, the Mucoromycotina class, with the Mucor mucedo group (0.19%) was detected. The percentage of mould and yeast-like fungi in the seaside air is shown in Figure 3.

3.5. The qualitative assessment of mould and yeast-like fungi in the air of seaside towns in 2018

Ascomycota (96.17%), Basidiomycota (1.21%) and Zygomycota (Fungi Incertae Sedis) (2.62%) were detected in the air samples collected in 2018 in the seaside towns of Hel, Puck, Gdynia, Sopot and Gdańsk-Brzeźno (Figure 4). In the Ascomycota phylum, four classes of fungi were found: Eurotiomycetes (60.07%), Dothideomycetes (4.60%), Saccharomycetes (27.26%) and Sordariomycetes (4.24%).

The genus isolated within the Eurotiomycetes class were: Penicillium (20.72%), Aspergillus (38.5%) and Trichophyton (0.85%). Within the Penicillium genus, Penicillium section Viridicata (11.87%) and Penicillium section Chrysogena (8.84%) were detected. Within the Aspergillus genus, the species: Aspergillus section Nigri (37.05%) and Aspergillus section Fumigati (1.45%) were isolated. Within the Trichophyton genus, the Trichophyton mentagrophytes species complex was detected (0.85%). In the Dothideomycetes class, there was Cladosporium genus, the Cladosporium herbarum species complex (4.31%), and the Alternaria genus, with the Alternaria alternata species complex (0.29%). In the Sordariomycetes class, the Stachobytris genus, with the Stachybotrys chartarum species  complex (4.24%) was found. In the Saccharomycetes class, the Candida genus was isolated (27.26%). In the Basidiomycota phylum, the Cystobasidiomycetes class, the Rhodotorula genus, the Rhodotorula mucilaginosa species (1.21%) was detected. In the Zygomycota phylum, the Mucoromycotina class, with the Mucor mucedo group (2.62%) was found. In 2018, potentially pathogenic and allergenic mould and yeast-like fungi were detected in the seaside air, such as: A. section Fumigati (1.45%), S. chartarum complex (4.24%) and C. albicans (27.26%). The species were not observed in the years 2014-2017. The percentage share of these mould and yeast-like fungi in the samples of seaside air is shown in Figure 4.

Discusion should not contain results like in L 311-320

Author’s Response: Thank you for your remark. As suggested by Reviewer 2, we have removed some of the text containing the results, such as L 311-320.New text is: L338-366

----------------------------------------------------------------------------------------

I encourage authors to less discuss negative effects of species like L321-329 and be more focused on their occurrence in environment especially in context of results like in L350-353.

Author’s Response: Thank you for your remark. As suggested by Reviewer 2, we removed some of the text related to the negative effects of species, such as L321-329. We have added a new text in which we describe the occurrence of mold in the environment, especially in the context of results such as L350-353.

New text is: L 388-401

----------------------------------------------------------------------------------------

Language quality and preparation of manuscript also have some flaws

Author’s Response: Thank you for your remark. As suggested by Reviewer 2, the publication was checked by a native speaker.

In table 1 p-value testu Freidmana comes from Polish

Author’s Response: Thank you for your remark. Table 1 has been deleted.  ---------------------------------------------------------------------------

L244 L266 L350 Genii should be genera

Author’s Response: Thank you for your remark. As suggested by Reviewer 2, we revised names from Genii to genera in L244, L266, L350.  ----------------------------------------------------------------------------------------

Latin names should be used in short form when appears second and next time in text

Author’s Response: Thank you for your remark. As suggested by Reviewer 2, we corrected the Latin names in the text and used the short form when they appeared for the second and subsequent time in the text.

----------------------------------------------------------------------------------------

I would encourage authors to do new statistical analysis, correct English, prepare new manuscript and resubmit it as the topic is interesting.

Author’s Response: As suggested by Reviewer 2, we conducted a new statistical analysis (PCA), and we prepared a new results and discussion.  The English language was improved by a native speaker.  We have done our best to ensure that the corrected version of the manuscript can be read clearly, and is concise and readable for the reader.
